# Intensive Care Unit Nurses’ Experiences in Caring for End-of-Life Patients in Saudi Arabia: A Qualitative Study

**DOI:** 10.3390/ijerph21070931

**Published:** 2024-07-17

**Authors:** Naif H. Alanazi

**Affiliations:** Medical-Surgical Department, College of Nursing, King Saud University, Riyadh 12372, Saudi Arabia; nalanazz@ksu.edu.sa or alanazinaif993@gmail.com

**Keywords:** end-of-life care, intensive care unit, nurse, qualitative, Saudi Arabia

## Abstract

**Background:** Although extensive research on appropriate treatments has been conducted, how nurses provide care to patients at the end-of-life (EOL) is unclear, particularly among intensive care unit (ICU) nurses in Saudi Arabia. **Purpose:** To explore intensive care unit nurses’ experiences in providing end-of-life care in Saudi Arabia. **Methods:** This study utilized an exploratory, descriptive, qualitative approach. A purposive sampling technique was used to recruit ICU nurses (n = 10) working in ICUs at a tertiary teaching hospital in Riyadh, Saudi Arabia. Using a semi-structured interview guide, ten individual interviews were conducted. Data were analyzed using thematic analysis. The trustworthiness of this study was ensured by following Lincoln and Guba’s (1985) criteria. **Findings:** Four major themes related to ICU nurses’ experiences of providing EOL care emerged, including: “feeling challenged but driven”, “holistic caring”, “collaborative working ethics”, and “caring for the undying and dying”. **Conclusions:** This study adds to the body of knowledge about the experience of ICU nurses caring for EOL patients. It offers valuable insights into challenges, coping strategies, holistic caring, collaboration, and the management of critical or dying patients at EOL in the ICUs.

## 1. Introduction

The primary objective of intensive care units (ICUs) is to stabilize and restore patients from a critical state to their prior state of health. Despite advancements in science and technologies, the trends of ICU death rates have remained high over 35 years (1985 to 2021), as reported in 210 ICU infection prevention studies [1]. As reported in a prospective multicenter study involving 96 adults ICUs in France, the rates of anticipated and unexpected patient deaths in ICUs are 68% and 32%, respectively [2]. The Centers for Disease Control [3] reported that the leading causes of death in ICUs are cancer, heart diseases, injuries, and respiratory infections. A hospital in Saudi Arabia reported a coronavirus-related death rate of 39% in its ICU [4]. ICUs are characterized by diverse and unpredictable environments where patients and families experience a prolonged and uncertain process of healing or the inevitable loss of a loved one. Severely or terminally ill patients in ICUs often undergo life-saving technical and medical therapies aimed at preventing death [5]. These patients receive ventilator support, cardiac resuscitation, and life-sustaining medication and therapies [6], and they are normally in a comatose state and incapable of communicating their needs [7]. Dying in an ICU is perceived as a sorrowful and unnatural experience in an unfamiliar setting with unfamiliar technology, especially when family members are not permitted to be by a patient’s bedside [8]. Thus, nurses in ICUs must plan end-of-life care (EOLC) for ICU patients and family members, despite the nurses’ primary goal being to save lives.

ICU nurses play an essential role in care delivery to critically ill patients and fully understand that delivering EOLC is a necessary and ethical duty [9]. By relying on patient-centered holistic treatments, this care simply enables patients to achieve their end-of-life preferences [10]. According to the National Institute for Health and Care Excellence [11], a quality standard for caring for death and dying patients and providing high-quality care improve patients’ quality of life by ensuring that they receive positive healthcare services in a safe environment. In 2014, the World Health Assembly encourages healthcare professionals, including ICU nurses, to develop palliative care capacity [12]. Therefore, palliative care is an essential part of intensive care for critically ill patients at the time of ICU admission, rather than a mutually restrictive option or a result of failed efforts for life-prolonging care.

In Saudi Arabia, nurses deliver EOLC in ICUs [13]. Terminally ill patients receive some form of care until they can go home, die in the ICU, or receive special care and treatment [14]. Brooks et al. [15], Langley et al. [16], and Badir et al. [17] reported that most ICU nurses from Australia, Turkey, and South Africa consider effective pain management and engagement with patients and families appropriate nursing strategies in EOLC. Several ICU nurses performed these interventions for dying patients and claimed that the patients they cared for died meaningful and peacefully [18].

However, although extensive research on appropriate treatments has been conducted, how nurses provide care to patients at the end of life is unclear [19,20]. The EOLC before an ICU death was reported by ICU nurses to be in the form of aggressive medical treatments in Saudi Arabia, with notable obstacles in its provision due to lack of EOLC awareness, language barriers, non-involvement of family in decision-making, cultural differences, and doctors who differed in their opinions [21]. Similar findings to Mani and Ibrahim’s [21] study were reported by ICU nurses in a qualitative study of Alshehri et al. [22], indicating that EOLC still faces challenges with family involvement in the care goals of patients and is determined by do not resuscitate (DNR) orders. Andersson et al. [23] found that nurses, even with necessary training and experience in transition care, still encounter challenges in providing care for terminally ill patients in surgical ICUs because of inadequate preparation. Lack of palliative care experience remains a concern among nurses, and support for patients and families is often insufficient [24] because of the shortage of ICU nurses [25]. Thus, providing the best possible care at the end of life is a challenge to ICU nurses [26]. ICU nurses are exposed to grief and human suffering [27], which induce anxiety and uncertainty on how to cope with procedures for handling a patient’s death [28], in addition to burnout and emotional exhaustion [29], especially during the COVID-19 pandemic [30]. Support and care for patients and families are essential during these difficult times. This study aimed to explore the lived experiences of ICU nurses in caring for dying or patients at the end of life in ICUs.

## 2. Materials and Methods

### 2.1. Design, Sample and Setting

This study aimed to examine the experiences of nurses who provide EOLC to patients in an ICU environment. This qualitative study adhered to the COnsolidated criteria for REporting Qualitative research (COREQ) guidelines Appendix A [31]. A phenomenological descriptive design was used. Phenomenology was selected because it allows researchers to perceive people’s life experience while considering their environments [32]; specifically, the actual experiences of ICU nurses implementing EOLC. After conveying the study design and describing the research environment, the researchers performed a series of steps to discover the phenomenon, including participant recruitment, data gathering and management, data analysis, and maintaining trustworthiness [32].

This study purposively recruited participants who were registered nurses and provided EOLC to patients and their families in ICUs. For phenomenological research such as the current study, Creswell [33] suggested 5–25 participants, whereas Morse [34] recommended six. The participants had current or recent experience in the ICUs of a government hospital in Saudi Arabia. Their professional standing and experience records contributed to the data’s credibility [35,36]. A non-probability convenience sampling method was used to recruit nurses from the ICUs who met the inclusion and exclusion criteria. The inclusion requirements were that the participants should be licensed ICU staff nurses working full time, currently working within the ICUs at the time of the interview, have at least two years of experience caring for dying patients and helping their families, willingly participated, and have provided consent prior to interviews. Participants who did not meet the inclusion criteria were excluded from the study.

### 2.2. Recruitment of the Participants

The recruitment process began through an orientation given to ICU nurse managers by the researcher to recruit possible eligible ICU nurses as participants based on the inclusion and exclusion criteria of the study. ICU nurse managers were given a copy of the inclusion and exclusion criteria, and they identified potential participants who met the criteria. ICU nurse managers recruited potential participants by sending an email invitation, participant information sheet, and informed consent form using their hospital email addresses. ICU nurse managers were only responsible for contacting potential participants and were not involved in screening ICU nurses who were eligible to participate or in collecting or keeping research data. The contact information of the researcher (i.e., email address and mobile number) used only for this research were included in the participant information sheet. ICU nurses who willingly volunteered to take part in the interview were requested to contact the researcher. In addition to obtaining written informed consent, the researcher secured verbal consent from each ICU nurse before the start of their interview. The researcher arranged interview dates and times that were convenient for each ICU nurse.

### 2.3. Data Collection Procedures

Semi-structured face-to-face interviews were conducted. The participants sat at a square table in a conference space. Recording was obtained with a digital audio recorder with participants’ permission. Participants were told that the entire interview would be performed in 40 min. The interviews varied in length from 30–90 min.

Two open-ended questions were employed as proposed by Forero et al. [35]:Can you tell us about your feelings? What it is like to take care of a dying patient and support their family in an ICU?Can you describe what type of care do you believe is appropriate for EOLC patients?

Interviews were performed to the point of data saturation, where no new information emerged.

### 2.4. Ethical Considerations

Ethical approval was reviewed by and obtained from an institutional review board in a medical city in Riyadh, Saudi Arabia. Written and verbal consent were also obtained from the participants. A face-to-face interview was digitally captured, transcribed, and archived in a secure location. The participants were told that their contributions would be kept private and that they could leave at any time of the interview. Each participant’s response was viewed as essential and considered significant. The sessions were designed to allow participants enough time to think about and focus on their experiences.

### 2.5. Data Analysis

The benefits of recording frequently occurring trends have been demonstrated by Kalu and Bwalya [37], Braun and Clarke [38] and Forbes [39]. This approach promotes the recognition of recurrent patterns and concepts and ensures that participant experiences are entirely included in the data analysis [40]. The recordings were transcribed, the records carefully read, and then the emerging themes were defined and coded per these recommendations. The data analysis procedure was based on the method proposed by Silver and Lewins [41]. NVivo 10.1 software was used by the researcher to manage the research data, and thematic analysis that adhered to Braun and Clarke’s [42] guidelines was used by the researcher to analyze the data obtained from the ICU nurses. The thematic analysis of the results was double-checked and confirmed by two qualitative research experts who are nurses who have a PhD degree in nursing and extensive experience in qualitative research methods.

### 2.6. Rigor

This study used the four standards of Lincoln and Guba [43] to assess the study’s rigor in terms of credibility, validity, dependability, and transferability. Researchers’ reputation was built by extended interaction with the evidence, observations, detection of dissimilar findings, and participant verification. This study employed ICU nurses with considerable experience in providing care to dying patients to collect a large and rich dataset. This study examined and explained each sample from sampling to data collection and interpretation and compared the findings. Two qualitative research experts double-checked the definitions and coding. The researchers used more than two questions to study the phenomenon and validate the observations. To strengthen dependability, coding schemes were used.

## 3. Results

In this study, a total of 10 ICU nurses were recruited to participate in semi-structured interviews (Table 1). Of the 10, eight were females and two were males. Seven were non-Saudis and three were Saudis. The age of the participants ranged from 29 to 48 years and their years of experience working in ICUs ranged between 3 and 21 years, with an average of 9.2 years. Nine out of the ten ICU nurses had training in palliative care or EOLC.

Four main themes emerged after thorough reviews and deliberations using the transcripts and results of the analysis. These themes comprised nine theme clusters. The four main themes illustrate the actual experience of nurses working in ICUs in this specific setting (Table 2).

### 3.1. Feeling Challenged but Driven

This major theme illustrates the experience of the nurses who encounter challenges and how they respond to these circumstances when working in ICUs. Three theme clusters emerged from this main theme.

#### 3.1.1. Subtheme 1.1: Experiencing Both Emotional and Caring Challenges

The ICU nurses described their motivation to deliver the best nursing care to their patients but also that they are challenged under certain circumstances. The ICU nurses wanted to provide all-out care because most patients are either critically or dying but were faced with undesirable circumstances, leading to frustrations. Several challenges can sway negative feelings, such as stress, hopelessness, and sadness. Thus, this theme cluster refers to the disputes that nurses usually encounter in ICUs when dealing with unfavorable circumstances and providing care to patients.


*Because still there is pain, they can see that there is discomfort. Or maybe they are thinking that we are just neglecting their patient. I would feel a little bit emotional for this one because you would feel sorry for them. I mean, some of the patients are very young. Some of them are very old. We think they can still do more things in their lives.*
[P5]

#### 3.1.2. Subtheme 1.2: Pandemic-Induced Changes

The ongoing pandemic worsens several existing challenges in ICUs, such as staff shortages and limited resources. The increasing influx of contagious patients and high caring demands due to the increasing complexity of diseases makes the work in ICU extremely challenging and risky for ICU nurses. An organization’s responses, including changing the nurse–patient ratio, stringent infection control, and no-vacation policy, makes the work frustrating and stressful to ICU nurses.


*So, I think that is the biggest issue nowadays, especially this pandemic. Because now we are mixing the ICU, there is a COVID-ICU and the Adult ICU.*
[P3]

*The stress is already more than our hands this pandemic. All the staff wanted to have their vacation. In some instances, especially when the unit was very busy and chaotic, we became emotional*.[P6]

#### 3.1.3. Subtheme 1.3: Developing Adaptive Coping Strategies

Although the ICU environment has become more challenging than it was before the pandemic, nurses are still motivated and optimistic that these negative developments will eventually end. The ICU nurses must become patient advocates, support their team members, and serve as a source of support for families. The nurses view diseases and degeneration as parts of human life’s journey, and thus they show readiness and acceptance toward the death of patients and show empathy to the bereaved families.


*Coping strategies should be ready if you know the patient is dying. You should ready yourself regarding the situation. You know what, in my ICU experience, sometimes I am getting used to it. By the experience, because if you encounter another patient like this, you will be ready regarding the situation.*
[P2]

### 3.2. Theme 2: Holistic Caring

Codes such as supportive care, end of life, and deathbed indicate that patients in ICUs do not have the same power and energy as other patients. The nurses’ holistic approach in caring for ICU patients hinges upon their psychological, spiritual, and physical well-being and policies.

#### 3.2.1. Subtheme 2.1: Providing Psychological, Physical, and Spiritual Support

ICU nurses value the needed health support for patients and families. Providing comfort measures, managing pain, and ensuring proper nutrition, support with ventilation, pharmacological interventions, infection control, and assisted self-care have been the forerunning activities of ICU nurses, assisted by their respective healthcare teams. ICU nurses support bereaved families by allowing them to pray for patients, particularly by *Qur’an* readings, even if nurses have different religious beliefs.


*Likely, for Arab, Quran reading is more important that is what I believe more important. That one, that I prefer. Then pain management, supportive measures, nutrition. All care, personal care, mouth care, like all care, total care we must provide for the patient.*
[P1]

#### 3.2.2. Subtheme 2.2: Ensuring Policy-Based Caring

The ICU nurses’ functions in delivering patient care are bounded by policies based on their license and organization. The policies state the limitations and scope of practice for healthcare workers. The ICU working environment follows these terms, rendering the ICU teams’ practices specific and defined.


*Because everything is by the policy, we must go through. However, if we must give painkiller, pain management, and supportive measures that we must give for all DNR patients or end-of-life care patients if what is written in the policy. They need to get. They have the right to get all supportive measures.*
[P10]

### 3.3. Theme 3: Collaborative Working Ethics

The working atmosphere in the ICU depicts teamwork and collaboration. Each healthcare profession works for a specific aim and scope to help patients recover or expire with dignity. However, nurses go beyond what is expected of them to ensure holistic and all-out care to patients and their families.

#### 3.3.1. Subtheme 3.1: Delineated Working Tasks

Doctors are in charge of assessing patients and planning and ordering treatments. The other healthcare professionals include dietitians, physiotherapists, respiratory therapists, physical therapists, and others. Each team member has a specific scope and aim of practice.


*We are in a multidisciplinary team. There are available nurses, available doctors all the time, RTs, pharmacist, dietitian. Sometimes they are also giving their thoughts and opinions regarding that patient. For dietitian, they are giving nutrition to that patient. The RTs, they are giving ventilator support and all. Then the doctors are always available anytime they need, the doctors, and their family if they want something like explanation regarding the patient’s diagnosis, they are always available.*
[P9]

#### 3.3.2. Subtheme 3.2: Providing Unconditional Care

ICU nurses are patient advocates with a positive attitude and are altruistic, especially when they ensure quality of care and safety. Aside from preparing and implementing care plans, they remind healthcare professionals about the policies and needed tasks. The unwavering care of nurses binds all teams involving patients and families to resolve problems, despite the shortcomings in ICUs.


*For me, as a nurse, any different kinds of patient, either dying or not. That is our profession. We cannot really choose what is to be taken care of, like the conscious patient or the dying patient. Both are what we are dealing with. For me, if I choose that one, I will choose this really, dying patient.*
[P7]

### 3.4. Theme 4: Caring for the Undying and Dying

This major theme is one of the most interesting. The ICU nurses explained that the care they provide in ICUs is not all about patients who will eventually die.

#### 3.4.1. Subtheme 4.1: Caring for the Undying

The ICU nurses corrected the misconception that patients in ICUs are all in the terminal stage. Nurses offer active care primarily to the conscious, young, and patients not indicated for EOLC care. These patients are commonly diagnosed with a disease that the nurses can effectively manage. Caring for the undying is perceived by the nurses as rewarding, especially when the patients recover and are ordered for transfer to hospital wards or discharged for recuperation at home.


*This is not palliative. Every patient there is different, and ICU means critical care. Critical patient means we are managing with them. I can give continuity of care and take care of all the patients. I can take care of the same patient continuously I can do. Not like end-of-life care because we are transferring the patient, discharging the patient. They will get better.*
[P4]

#### 3.4.2. Subtheme 4.2: Doing Palliative Care

The ICU nurses expect that a patient is dying mainly when the orders made by doctors include DNR code or EOLC. Nurses offer all care and privacy even to patients who are not subjected to a DNR order to preserve human dignity.


*We are just giving supportive care like we cannot do anything for them but let them die holistically. Because the patient has an end-of-life treatment already. So, it is not easy for dying patients, to take care of them, especially to the family.*
[P8]

## 4. Discussion

This study’s findings revealed that ICU nurses’ experiences in caring for critically ill patients at their EOL were focused on the following major themes: feeling challenged but driven, holistic caring, collaborative working ethics, and caring for the undying or dying. The findings of the present study are comparable to a qualitative study that used a classic Grounded Theory approach among 22 hospice nurses in UK [44]. Interestingly, the core category or theme in Griffith and Gelling’s [44] study is about ‘the shared ideal’, based on one of the instances of giving good and holistic care to dying patients, which is consistent with the present study findings. Hence, Griffith and Gelling [44] successfully came up with a meaningful substantive theory based on the findings of their study that distinguished hospice nurses as having a strong sense of identity with their colleagues who shared their ideal ways of providing EOLC for dying patients. This strong sense of identity at work gave hospice nurses the feeling of being prepared to provide EOLC [44].

For the first major theme, the ICU nurses described that the challenges in ICUs were attributed to shortcomings either beyond their control or due to events that could be mitigated with adaptive coping strategies. ICU nurses expressed burnout, accentuated recently by sudden changes in ICUs during the pandemic, such as the no-vacation policy and extensions of shift work. According to AlMubark et al. [45], the burnout experience in ICUs already existed before the pandemic, and hospital organizations must address this problem to ensure a robust healthcare workforce in ICU settings. Several nurses in ICUs in Saudi Arabia have experienced moderate or high levels of exhaustion or burnout before [46] and during [47,48] the pandemic. Alzailai et al. [47] reported that ICU nurses felt stressed, and feared being infected and considered it as a torture when caring for critically ill patients during the pandemic because they were always exposed to the virus and some of their colleagues died. In this study, some ICU nurses explained that they want to leave the ICUs because of stress and the increasing workload during the pandemic. Therefore, healthcare authorities need to establish mental health support services, increase the workforce, incorporate more supervision into practice, and explore alternative work patterns for nurses and other healthcare workers to avoid staff shortages. Moreover, the ICU nurses expressed that the way to cope with challenges in ICUs is to establish adaptive coping strategies. This finding is consistent with Buheji and Buhait’s [49] study indicating that nurses can cope with challenges by directing coping toward reducing the complexity of a problem, increasing information processing, mitigating side effects, and analyzing tasks. This strategy was indicated in the interviews of the ICU nurses whenever they establish support and coping. For instance, they support the value of teamwork because it reduces workload and builds the value of relationships among the members of the healthcare team. In addition, the ICU nurses experienced a sense of satisfaction when they brought comfort to patients and family members despite the challenges they encountered in ICUs during the pandemic.

In the second major theme, the provision of holistic care in ICUs has also been explored by Albaqawi et al. [50] in Saudi Arabia, with consistent findings in offering psychological, spiritual, and physical support to patients. ICU nurses acknowledge support from fellow nurses, other healthcare professionals, and patients’ families. For families, the ICU nurses allow patients’ relatives to pray for their spiritual health and be informed of their status, especially prognoses. According to Albaqawi et al. [50], ICU nurses in Saudi Arabia must adhere to the core Islamic tenets because it can help build a positive attitude among ICU patients. Hence, the awareness of personal and spiritual beliefs systems of ICU nurses is essential to stress management and caring challenges. In physical health, pain has been the most common experience that ICU nurses encountered when they cared for ill and dying patients. This unpleasant feeling has also been reported by ICU nurses in Iran [51]. In this study, the ICU nurses showed concern about a patient’s pain and asked doctors to administer pain killers. In relation to psychological support, EOLC has a profound impact not only on patients but also on their families and care providers [52]. In this study, ICU nurses stated that they feel emotional and sad, especially when they care for young critically ill patients or dying patients, and are empathetic to bereaved family members. Hence, the ICU nurses involved families in their care plans. Additionally, doctors need to update families about the prognoses and treatments of their patients. However, evidence has shown that Saudi families were less involved in care or lacked support because of the high risk of COVID-19 infection [53]. Nevertheless, ICU nurses still exert efforts to involve families through telecommunication. Moreover, the ICU nurses described their caring experience as aligned with the policies of hospitals and the ministry of health. This finding is consistent with the study results of Arabi et al. [54] indicating that ICUs must be prepared for the influx of patients requiring intensive care. Thus, healthcare professionals should review their models, triage, and EOLC principles according to the current need for changes [54]. These changes require the commitment of healthcare professionals in planning and sustaining improvements in healthcare systems, including ICUs.

For the third major theme, collaborative working ethics, ICU nurses play an essential role in providing family care and collaborating with ICU teams [55]. In the present study, the nurses described the ICU environment as “working together”. The ICU nurses perceived themselves as the link of patients and families to the rest of the healthcare team. According to Ervin et al. [56], ICUs provide care to severely ill patients; thus, interprofessional collaboration is vital to providing optimal care. The ICU nurses reiterated supportive care in the interviews as an approach for managing patients with the assistance of a team. Shortcomings in performing each other’s roles exist, but ICU nurses remind their teams about the stipulations of and adherence to policies. For instance, nurses are not allowed to explain diagnoses and treatments. They remind doctors to accomplish such tasks to patients and family members. In ICU settings, team dynamics should be advanced, and actionable intervention can help a team to deliver optimal care. In addition, ICU nurses described working in ICUs as delivering unconditional care as they deal with death, dying, morbidity, and conflict. They described unconditional care as providing unwavering care that binds or unifies the healthcare team in rendering EOLC in ICUs. Having a large ICU team requires good collaborative and communication skills, which are essential to the delivery of quality EOLC in ICUs [52]. However, providing unconditional EOLC for dying patients could be challenged by a dilemma related to cultural norms on the prevention of death justifying the utilization of a DNR order while concurrently providing life-saving medical interventions and the obstacles of supporting families when their patient is at EOL in the ICU [22].

The last major theme was the experiences of ICU nurses that evolved in providing palliative care, both for undying patients (those who recovered from their critical conditions) and those at EOL. The ICU nurses explained that they provide all-out unwavering care to two types of patients: the critically ill and the dying. Critical care is mainly performed for patients who may recover. The nurses felt that the experience is rewarding when these patients recover and are sent home for discharge. Meanwhile, palliative patients are often indicated with DNR order or at times EOLC. For these patients, ICU nurses still offer a holistic approach to caring. The challenge with palliative care is when a family cannot accept the death of a patient. According to Kia et al. [51], bereaved families described the loss of loved ones in ICUs as grief and related how ICU events affected their bereavement. End-of-life decisions elicit strong feelings from healthcare professionals. Laurent et al. [57] explained that nurses’ feelings toward their patients influence decision-making in the end-of-life decision-making process. Similarly, physicians’ feelings toward their patients’ families influence their decisions. Although ICU professionals have well acknowledged their emotions, the understanding of these emotions remains lacking and is often overlooked. Thus, they often make inefficient decisions. In this study, the ICU nurses described that doctors must be reminded about communicating with families about poor prognoses to lessen the burden caused by a patient’s death. According to Tripathy et al. [58], ICU nurses in India play a crucial role in EOL discussions. ICU nurses can be involved in decision-making because this approach may improve their well-being and the care they provide to patients and families. Thus, nurses as well as doctors should work with families to make decisions that can offer optimal care plans to ICU patients.

This study was limited to the shared experiences of, and to a relatively small sample of, ICU nurses in the selected setting and did not involve any direct perspectives from other members of healthcare teams or organizations, which limits the generalizability of these findings.

## 5. Conclusions

This study’s findings depict the context and experiences of nurses in one of the ICUs in Saudi Arabia. The challenges experienced by the ICU nurses are related to caring, emotions, organization, and family involvement. Being in an ICU setting is interpreted by the ICU nurses as working together with healthcare professionals with different specialties who deliver all-out care to either critical or dying patients. The nurses refer to the holistic approach of caring for ICU patients as caring across multiple dimensions, such as psychological, physical, spiritual, and organizational. Therefore, this study adds to the body of knowledge about the experience of ICU nurses caring for EOL patients. It offers valuable insights into challenges, coping strategies, holistic caring, collaboration, and the management of critical or dying patients.

Establishing bereavement programs can help the families of dying patients to go through the grieving process. Moreover, healthcare professionals are encouraged to continue participating in stress management programs to build their capacity to adapt to changes and their resilience for mitigating undesirable circumstances. Owing to the COVID-19 restrictions, both programs can be performed via online platforms, such as Zoom or Microsoft Teams. Counseling or a peer facilitator group can address the feeling of burnout and intent to leave ICUs or organizations because of an increasing workload due to the current pandemic. Organizational support plays a crucial role when patient influx and frequency of deaths increase in ICUs. Enhanced communications with teams is essential, and a reflective learning program can be established to build relationships and advance interprofessional collaboration. Lastly, an organization’s professional development should be prioritized for ICU nurses, focusing on organization development to change, cultural competence, strategic management, and decision-making, which are necessary to the delivery of optimal care in ICU settings.

## Figures and Tables

**Table 1 ijerph-21-00931-t001:** Demographic profile of the ICU nurses.

Participants	Gender	Age (In Years)	Nationality	Number of Years Working in ICU	Having Training in Palliative Care or EOLC
P1	Female	29	Saudi	3	No
P2	Female	35	Filipino	8	Yes
P3	Male	31	Filipino	5	Yes
P4	Female	36	Indian	9	Yes
P5	Female	34	Jordanian	7	Yes
P6	Female	40	Indian	11	Yes
P7	Female	38	Saudi	6	Yes
P8	Female	45	Filipino	15	Yes
P9	Male	36	Saudi	7	Yes
P10	Female	48	Indian	21	Yes

**Table 2 ijerph-21-00931-t002:** Main themes and subthemes.

Subthemes	Themes
Experiencing both emotional and caring challenges Pandemic-induced changes Developing adaptive coping strategies	Feeling challenged but driven
Providing psychological, physical, and spiritual support Ensuring policy-based caring	Holistic caring
Delineated working tasks Providing unconditional care	Collaborative working ethics
Caring for the undying Doing palliative care	Caring for the undying and dying

## Data Availability

The data presented in this study are available on request from the corresponding author.

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
