# Peer review of "Intensive Care Unit Nurses’ Experiences in Caring for End-of-Life Patients in Saudi Arabia: A Qualitative Study"

_ijerph, 2024, doi:10.3390/ijerph21070931_

Round 1

Reviewer 1 Report

Comments and Suggestions for Authors

The impact of the manuscript is limited by the small number of ICU nurses who were interviewed.  However, even with that limitation, it provides useful insight into the challenges experienced by ICU nurses when providing end of life care, particularly during the height of COVID-19 pandemic.  The abstract, introduction, materials and methods and results sections are well written and informative.  However, the discussion is rambling and should focus more closely on the four themes identified and could be significantly more brief.  The limitations section should state that the small number of subjects is a significant limitation.  

Author Response

Reviewer 1 Comments

Comment 1: The impact of the manuscript is limited by the small number of ICU nurses who were interviewed.  However, even with that limitation, it provides useful insight into the challenges experienced by ICU nurses when providing end of life care, particularly during the height of COVID-19 pandemic.  The abstract, introduction, materials and methods and results sections are well written and informative. 

Response 1: I agree with the honorable reviewer and thank you so much for your favorable comments.

Comment 2: However, the discussion is rambling and should focus more closely on the four themes identified and could be significantly more brief. 

Response 2: I also agree. With this, I revised the discussion section in an organized manner according to the four themes and in a briefer version. Correspondingly, some citations were deleted in the Discussion section as well as the references in the References section.

Comment 3: The limitations section should state that the small number of subjects is a significant limitation. 

Response 3: Agree and added this in the limitations section. Again, thank you so much for your helpful comments and hope that the revised version of my paper is acceptable to you. May this also help the editor in deciding to accept my paper for publication in the IJERPH journal.

Reviewer 2 Report

Comments and Suggestions for Authors

Thank you for the opportunity you have given me

The article is well-written and addresses an important topic in nursing, please

The approach of this study is conventional content analysis.

 Why haven't researchers used triangulation to increase data reliability?

The demographic information table should be attached.

Author Response

Reviewer 2 Comments

Comment 1: Thank you for the opportunity you have given me

Response 1: You are most welcome and thank you too for your helpful comments.

Comment 2: The article is well-written and addresses an important topic in nursing, please

Response 2: I gladly agree and thank you once again.

Comment 3: The approach of this study is conventional content analysis.

Response 3: May I clarify this comment with the honorable reviewer as this qualitative study employed a phenomenological descriptive design and used thematic analysis. I did not revise anything in my manuscript based on this comment.

Comment 4: Why haven't researchers used triangulation to increase data reliability?

Response 4: Data triangulation was not used in this study, but I used investigator triangulation as I consulted two qualitative research experts to double-check the coding and themes, as indicated in section 2.5 Rigor.

Comment 5: The demographic information table should be attached.

Response 5: I adhere to this comment and provided the demographic information in the beginning of the Results section as Table 1. Correspondingly, Table 1 Main Themes and Subthemes has been revised as Table 2.

Reviewer 3 Report

Comments and Suggestions for Authors

The article addresses an important topic of research dedicated to a better understanding of the process of caring for dying patients within acute departments (namely Intensive Care Units), focusing on the role of nurses and their experience. The paper adopts a qualitative approach, which is correctly carried out. Nonetheless, I find some flaws that need to be addressed. Below are my main comments.

1.      It is unclear whether the nurses involved in the study had received previous training in palliative care. This should be an important feature of the approach, especially in the provision of EOLC.

2.      The verbatim related to the subtheme 4.1 Doing Palliative Care it is not clear to me, especially the two following sentences: “let them die holistically” and “it is not easy for dying patients to take care of them because aside from, especially to the family”. Not all dying patients need palliative care and conversely not all the patients receiving palliative care necessarily die. Moreover, holistic approach (worrying for all the aspects/needs involved in a caring process), should be applied both to dying/undying patient: the shift shouldn’t be in the goals of the assistance?

3.      Consistently with an insight pointed out in the introduction section (lines 43-45 p. 1) I would elaborate on the differences in the contents of the caring in subtheme 3.2 (for instance, more on spiritual side or psychological aspects for the dying patients, more involvement of the family with end-stage patients, and so on). Are there any dilemmas, professional dilemmas, with different patients/conditions? How nurses cope with those dilemmas?

4.      Author refers interestingly to studies about the impact of pandemic on ICUs’ nurses’ burnout. A reference would be useful to better understand the reverberations on critical patients.

5.      Author refers to a study (Mani, & Ibrahim, 2017) pointing out that “Care before an ICU death is an uncommon occurrence in Saudi Arablia” (line 67, p. 2). I would suggest to mitigate the expression or to find stronger evidence: the authors of that study themselves, indeed, declared limitations in generalizability of their findings (The findings of this study cannot generalized because it was conducted in one health care setting with a relatively small sample size. p. 719). Furthermore, the paper mentioned focuses more on analyzing the obstacles in providing EOLC in ICUs than assessing penetration rate of EOLC in ICU.

Author Response

Reviewer 3 Comments

Comment 1: The article addresses an important topic of research dedicated to a better understanding of the process of caring for dying patients within acute departments (namely Intensive Care Units), focusing on the role of nurses and their experience. The paper adopts a qualitative approach, which is correctly carried out. Nonetheless, I find some flaws that need to be addressed. Below are my main comments.

Response 1: Thank you very much for your valuable comments that helped in the improvement of my revised manuscript. May my revisions based on your comments aid the academic editor in accepting my paper for publication.

Comment 2: It is unclear whether the nurses involved in the study had received previous training in palliative care. This should be an important feature of the approach, especially in the provision of EOLC.

Response 2: As also suggested by another reviewer, I provided this information together with other demographic profile of the ICU nurses  in the beginning of the Results section and 9 out of the 10 participants had palliative care or EOLC training.

Comment 3: The verbatim related to the subtheme 4.1 Doing Palliative Care it is not clear to me, especially the two following sentences: “let them die holistically” and “it is not easy for dying patients to take care of them because aside from, especially to the family”. Not all dying patients need palliative care and conversely not all the patients receiving palliative care necessarily die. Moreover, holistic approach (worrying for all the aspects/needs involved in a caring process), should be applied both to dying/undying patient: the shift shouldn’t be in the goals of the assistance?

Response 3: I agree with the honorable reviewer, and it has been indicated in Section 3.4.2. Subtheme 4.2: Doing palliative care, that: “Nurses offer all care and privacy even to patients who are not subjected to DNR order to preserve human dignity.” If I did not get what you meant exactly based on your comment, may I ask for further clarification and I am very willing to comply further and will do necessary or additional revisions.

Comment 4: Consistently with an insight pointed out in the introduction section (lines 43-45 p. 1) I would elaborate on the differences in the contents of the caring in subtheme 3.2 (for instance, more on spiritual side or psychological aspects for the dying patients, more involvement of the family with end-stage patients, and so on). Are there any dilemmas, professional dilemmas, with different patients/conditions? How nurses cope with those dilemmas?

Response 4: Thank you for this comment and I provided a revision based on this in the Discussion section, at the last part of the third paragraph. Please see Lines 396-400.

Comment 5: Author refers interestingly to studies about the impact of pandemic on ICUs’ nurses’ burnout. A reference would be useful to better understand the reverberations on critical patients.

Response 5: I agree and because of this comment, I provided a citation (Alzailai et al., 2023) explaining or presenting the impacts of the pandemic on ICU nurses’ burnout and stress. Please refer to this: Alzailai et al. (2023) reported that ICU nurses felt stressed, feared of being infected and considered it as a torture when caring for critically ill patients during the pandemic because they were always exposed to the virus and some of their colleagues died.”

Comment 6: Author refers to a study (Mani, & Ibrahim, 2017) pointing out that “Care before an ICU death is an uncommon occurrence in Saudi Arablia” (line 67, p. 2). I would suggest to mitigate the expression or to find stronger evidence: the authors of that study themselves, indeed, declared limitations in generalizability of their findings (The findings of this study cannot generalized because it was conducted in one health care setting with a relatively small sample size. p. 719). Furthermore, the paper mentioned focuses more on analyzing the obstacles in providing EOLC in ICUs than assessing penetration rate of EOLC in ICU.

Response 6: Thank you for this comment and I revised this as suggested. I also added a relevant citation by another study in Saudi Arabia (Alshehri et al., 2022). Please see in Lines 67-74.

Round 2

Reviewer 1 Report

Comments and Suggestions for Authors

This version of the manuscript is much improved.

Author Response

COMMENT 1: This version of the manuscript is much improved.

RESPONSE 1: Thank you very much for your positive comment.